# Statistical Cost Sharing

**Eric Balkanski**
Harvard University
ericbalkanski@g.harvard.edu

**Umar Syed**
Google NYC
usyed@google.com

**Sergei Vassilvitskii**
Google NYC
sergeiv@google.com

## Abstract

We study the cost sharing problem for cooperative games in situations where the cost function $C$ is not available via oracle queries, but must instead be learned from samples drawn from a distribution, represented as tuples $(S, C(S))$, for different subsets $S$ of players. We formalize this approach, which we call STATISTICAL COST SHARING, and consider the computation of the core and the Shapley value. Expanding on the work by Balcan et al. [2015], we give precise sample complexity bounds for computing cost shares that satisfy the core property with high probability for any function with a non-empty core. For the Shapley value, which has never been studied in this setting, we show that for submodular cost functions with bounded curvature $\kappa$ it can be approximated from samples from the uniform distribution to a $\sqrt{1-\kappa}$ factor, and that the bound is tight. We then define statistical analogues of the Shapley axioms, and derive a notion of statistical Shapley value and that these can be approximated arbitrarily well from samples from any distribution and for any function.

## 1 Introduction

The cost sharing problem asks for an equitable way to split the cost of a service among all of the participants. Formally, there is a cost function $C$ defined over all subsets $S \subseteq N$ of a ground set of elements, or *players*, and the objective is to fairly divide the cost of the ground set $C(N)$ among the players. Unlike traditional learning problems, the goal here is not to predict the cost of the service, but rather learn which ways of dividing the cost among the players are equitable.

Cost sharing is central to cooperative game theory, and there is a rich literature developing the key concepts and principles to reason about this topic. Two popular cost sharing concepts are the *core* [Gillies, 1959], where no group of players has an incentive to deviate, and the *Shapley value* [Shapley, 1953], which is the unique vector of cost shares satisfying four natural axioms.

While both the core and the Shapley value are easy to define, computing them poses additional challenges. One obstacle is that the computation of the cost shares requires knowledge of costs in myriad different scenarios. For example, computing the exact Shapley value requires one to look at the marginal contribution of a player over *all possible subsets* of others. Recent work [Liben-Nowell et al., 2012] shows that one can find approximate Shapley values for a restricted subset of cost functions by looking at the costs for polynomially many specifically chosen subsets. In practice, however, another roadblock emerges: one cannot simply query for the cost of an arbitrary subset. Rather, the subsets are passively observed, and the costs of unobserved subsets are simply unknown. We share the opinion of Balcan et al. [2016] that the main difficulty with using cost sharing methods in concrete applications is the information needed to compute them.

Concretely, consider the following cost sharing applications.

**Attributing Battery Consumption on Mobile Devices.** A modern mobile phone or tablet is typically running a number of distinct apps concurrently. In addition to foreground processes, a lot of activity may be happening in the background: email clients may be fetching new mail, GPS may be active for geo-fencing applications, messaging apps are polling for new notifications, and so on. All of these activities consume power; the question is how much of the total battery consumption should be attributed to each app? This problem is non-trivial because the operating system induces cooperation between apps to save battery power. For example there is no need to activate the GPS sensor twice if two different apps request the current location almost simultaneously.

**Understanding Black Box Learning** Deep neural networks are prototypical examples of black box learning, and it is almost impossible to tease out the contribution of a particular feature to the final output. Particularly in situations where the features are binary, cooperative game theory gives a formal way to analyze and derive these contributions. While one can evaluate the objective function on any subset of features, deep networks are notorious for performing poorly on certain out of sample examples [Goodfellow et al., 2014, Szegedy et al., 2013], which may lead to misleading conclusions when using traditional cost sharing methods.

We model these cost sharing questions as follows. Let $N$ be the set of possible players (apps or features), and for a subset $S \subseteq N$, let $C(S)$ denote the cost of $S$. This cost represents the total power consumed over a standard period of time, or the rewards obtained by the learner. We are given ordered pairs $(S_1, C(S_1)), (S_2, C(S_2)), \ldots, (S_m, C(S_m))$, where each $S_i \subseteq N$ is drawn independently from some distribution $\mathcal{D}$. The problem of STATISTICAL COST SHARING asks to look for reasonable cost sharing strategies in this setting.

## 1.1 Our results

We build on the approach from Balcan et al. [2015], which studied STATISTICAL COST SHARING in the context of the core, and assume that only partial data about the cost function is observed. The authors showed that cost shares that are likely to respect the core property can be obtained for certain restricted classes of functions. Our main result is an algorithm that generalizes these results for *all* games where the core is non-empty and we derive sample complexity bounds showing exactly the number of samples required to compute cost shares (Theorems 1 and 2). While the main approach of Balcan et al. [2015] relied on first learning the cost function and then computing cost shares, we show how to proceed directly, computing cost shares without explicitly learning a good estimate of the cost function. This high level idea was independently discovered by Balcan et al. [2016]; our approach here greatly improves the sample complexity bounds, culminating in a result logarithmic in the number of players. We also show that approximately satisfying the core with probability one is impossible in general (Theorem 3).

We then focus on the Shapley value, which has never been studied in the STATISTICAL COST SHARING context. We obtain a tight $\sqrt{1-\kappa}$ multiplicative approximation of the Shapley values for submodular functions with bounded curvature $\kappa$ over the uniform distribution (Theorems 4 and 11), but show that they cannot be approximated by a bounded factor in general, even for the restricted class of coverage functions, which are learnable, over the uniform distribution (Theorem 5). We also introduce a new cost sharing method called *data-dependent Shapley value* which is the unique solution (Theorem 6) satisfying four natural axioms resembling the Shapley axioms (Definition 7), and which can be approximated arbitrarily well from samples for any bounded function and any distribution (Theorem 7).

## 1.2 Related work

There are two avenues of work which we build upon. The first is the notion of cost sharing in cooperative games, first introduced by Von Neumann and Morgenstern [1944]. We consider the Shapley value and the core, two popular solution concepts for cost-sharing in cooperative games. The Shapley value [Shapley, 1953] is studied in algorithmic mechanism design [Anshelevich et al., 2008, Balkanski and Singer, 2015, Feigenbaum et al., 2000, Moulin, 1999]. For applications of the Shapley value, see the surveys by Roth [1988] and Winter [2002]. A naive computation of the Shapley value of a cooperative game would take exponential time; recently, methods for efficiently approximating

the Shapley value have been suggested [Bachrach et al., 2010, Fatima et al., 2008, Liben-Nowell et al., 2012, Mann, 1960] for some restricted settings.

The core, introduced by Gillies [1959], is another well-studied solution concept for cooperative games. Bondareva [1963] and Shapley [1967] characterized when the core is non-empty. The core has been studied in the context of multiple combinatorial games, such as facility location Goemans and Skutella [2004] and maximum flow Deng et al. [1999]. In cases with no solutions in the core or when it is computationally hard to find one, the balance property has been relaxed to hold approximately [Devanur et al., 2005, Immorlica et al., 2008]. In applications where players submit bids, cross-monotone cost sharing, a concept stronger than the core that satisfies the group strategy proofness property, has attracted a lot of attention [Immorlica et al., 2008, Jain and Vazirani, 2002, Moulin and Shenker, 2001, Pál and Tardos, 2003]. We note that these applications are sufficiently different from the ones we are studying in this work.

The second is the recent work in econometrics and computational economics that aims to estimate critical concepts directly from a limited data set, and reason about the sample complexity of the computational problems. Specifically, in all of the above papers, the algorithm must be able to query or compute $C(S)$ for an arbitrary set $S \subseteq N$. In our work, we are instead given a collection of samples from some distribution; importantly the algorithm does not know $C(S)$ for sets $S$ that were not sampled. This approach was first introduced by Balcan et al. [2015], who showed how to compute an approximate core for some families of games. Their main technique is to first learn the cost function $C$ from samples and then to use the learned function to compute cost shares. The authors also showed that there exist games that are not PAC-learnable but that have an approximate core that can be computed. Independently, in recent follow up work, the authors showed how to extend their approach to compute a probably approximate core for all games with a non-empty core, and gave weak sample complexity bounds [Balcan et al., 2016]. We improve upon their bounds, showing that a logarithmic number of samples suffices when the spread of the cost function is bounded.

## 2 Preliminaries

A *cooperative game* is defined by an ordered pair $(N, C)$, where $N$ is the ground set of *elements*, also called *players*, and $C : 2^N \to \mathbb{R}_{\geq 0}$ is the *cost function* mapping each *coalition* $S \subseteq N$ to its cost, $C(S)$. The ground set of size $n = |N|$ is called the *grand coalition* and we denote the elements by $N = \{1, \dots, n\} = [n]$. We assume that $C(\emptyset) = 0$, $C(S) \geq 0$ for all $S \subseteq N$, and that $\max_S C(S)$ is bounded by a polynomial in $n$, which are standard assumptions. We will slightly abuse notation and use $C(i)$ instead of $C(\{i\})$ for $i \in N$ when it is clear from the context.

We recall three specific classes of functions. *Submodular* functions exhibit the property of diminishing returns: $C_S(i) \geq C_T(i)$ for all $S \subseteq T \subseteq N$ and $i \in N$ where $C_S(i)$ is the marginal contribution of element $i$ to set $S$, i.e., $C_S(i) = C(S \cup \{i\}) - C(S)$. *Coverage* functions are the canonical example of submodular functions. A function is coverage if it can be written as $C(S) = |\cup_{i \in S} T_i|$ where $T_i \subseteq U$ for some universe $U$. Finally, we also consider the simple class of additive functions, such that $C(S) = \sum_{i \in S} C(i)$.

A *cost allocation* is a vector $\psi \in \mathbb{R}^n$ where $\psi_i$ is the *share* of element $i$. We call a cost allocation $\psi$ *balanced* if $\sum_{i \in N} \psi_i = C(N)$. Given a cooperative game $(N, C)$ the goal in the cost sharing literature is to find "desirable" balanced cost allocations. Most proposals take an axiomatic approach, defining a set of axioms that a cost allocation should satisfy. These lead to the concepts of Shapley value and the core, which we define next. A useful tool to describe and compute these cost sharing concepts is permutations. We denote by $\sigma$ a uniformly random permutation of $N$ and by $S_{\sigma<i}$ the players before $i \in N$ in permutation $\sigma$.

### 2.1 The core

The core is a balanced cost allocation where no player has an incentive to deviate from the grand coalition — for any subset of players the sum of their shares does not cover their collective cost.

**Definition 1.** *A cost allocation $\psi$ is in the* core *of function $C$ if the following properties are satisfied:*

- **Balance:** $\sum_{i \in N} \psi_i = C(N)$,
- **Core property:** *for all $S \subseteq N$, $\sum_{i \in S} \psi_i \leq C(S)$.*

The core is a natural cost sharing concept. For example, in the battery blame scenario it translates to the following assurance: No matter what other apps are running concurrently, an app is never blamed for more battery consumption than if it were running alone. Given that app developers are typically business competitors, and that a mobile device's battery is a very scarce resource, such a guarantee can rather neatly avoid a great deal of finger-pointing. Unfortunately, for a given cost function $C$ the core may not exist (we say the core is empty), or there may be multiple (or even infinitely many) cost allocations in the core. For submodular functions $C$, the core is guaranteed to exist and one allocation in the core can be computed in polynomial time. Specifically, for any permutation $\sigma$, the cost allocation $\psi$ such that $\psi_i = C(S_{\sigma<i} \cup \{i\}) - C(S_{\sigma<i})$ is in the core.

## 2.2 The Shapley value

The Shapley value provides an alternative cost sharing method. For a game $(N, C)$ we denote it by $\phi^C$, dropping the superscript when it is clear from the context. While the Shapley value may not satisfy the core property, it satisfies the following four axioms:

- **Balance:** $\sum_{i \in N} \phi_i = C(N)$.
- **Symmetry:** For all $i, j \in N$, if $C(S \cup \{i\}) = C(S \cup \{j\})$ for all $S \subseteq N \setminus \{i, j\}$ then $\phi_i = \phi_j$.
- **Zero element:** For all $i \in N$, if $C(S \cup \{i\}) = C(S)$ for all $S \subseteq N$ then $\phi_i = 0$.
- **Additivity:** For two games $(N, C_1)$ and $(N, C_2)$ with the same players, but different cost functions $C_1$ and $C_2$, let $\phi^1$ and $\phi^2$ be the respective cost allocations. Consider a new game $(N, C_1 + C_2)$, and let $\phi'$ be the cost allocation for this game. Then for all elements, $i \in N$, $\phi'_i = \phi_i^1 + \phi_i^2$.

Each of these axioms is natural: balance ensures that the cost of the grand coalition is distributed among all of the players. Symmetry states that two identical players should have equal shares. Zero element verifies that a player that adds zero cost to any coalition should have zero share. Finally, additivity just confirms that costs combine in a linear manner. It is surprising that the set of cost allocations that satisfies all four axioms is unique. Moreover, the Shapley value $\phi$ can be written as the following summation:

$$\phi_i = \mathbb{E}_{\sigma}[C(S_{\sigma<i} \cup \{i\}) - C(S_{\sigma<i})] = \sum_{S \subseteq N \setminus \{i\}} \frac{|S|!(n - |S| - 1)!}{n!}(C(S \cup \{i\}) - C(S)).$$

This expression is the expected marginal contribution $C(S \cup \{i\}) - C(S)$ of $i$ over a set of players $S$ who arrived before $i$ in a random permutation of $N$. As the summation is over exponentially many terms, the Shapley value generally cannot be computed exactly in polynomial time. However, several sampling approaches have been suggested to approximate the Shapley value for specific classes of functions Bachrach et al. [2010], Fatima et al. [2008], Liben-Nowell et al. [2012], Mann [1960].

## 2.3 Statistical cost sharing

With the sole exception of Balcan et al. [2015], previous work in cost-sharing critically assumes that the algorithm is given oracle access to $C$, i.e., it can query, or determine, the cost $C(S)$ for any $S \subseteq N$. In this paper, we aim to (approximately) compute the Shapley value and other cost allocations from *samples*, without oracle access to $C$, and with a number of samples that is polynomial in $n$.

**Definition 2.** *Consider a cooperative game with players $N$ and cost function $C$. In the* STATISTICAL COST SHARING *problem we are given pairs* $(S_1, C(S_1)), (S_2, C(S_2)), \ldots, (S_m, C(S_m))$ *where each $S_i$ is drawn i.i.d. from a distribution $\mathcal{D}$ over $2^N$. The goal is to find a cost allocation $\psi \in \mathbb{R}^n$.*

In what follows we will often refer to an individual $(S, C(S))$ pair as a *sample*. It is tempting to reduce STATISTICAL COST SHARING to classical cost sharing by simply collecting enough samples to use known algorithms. For example, Liben-Nowell et al. [2012] showed how to approximate the Shapley value with polynomially many queries $C(S)$. However, if the distribution $\mathcal{D}$ is not aligned with these specific queries, which is the case even for the uniform distribution, emulating these

algorithms in our setting requires exponentially many samples. Balcan et al. [2015] showed how to instead first learn an approximation to $C$ from the given samples and then compute cost shares for the learned function, but their results hold only for a limited number of games and cost functions $C$. We show that a more powerful approach is to compute cost shares directly from the data, without explicitly learning the cost function first.

## 3 Approximating the Core from Samples

In this section, we consider the problem of finding cost allocations from samples that satisfy relaxations of the core. A natural approach to this problem is to first learn the underlying model, $C$, from the data and to then compute a cost allocation for the learned function. As shown in Balcan et al. [2015], this approach works if $C$ is PAC-learnable, but there exist functions $C$ that are not PAC-learnable and for which a cost allocation that approximately satisfies the core can still be computed. The main result of this section shows that a cost allocation that approximates the core property can be computed from samples for *any* function with a non-empty core. We first give a sample complexity bound that is linear in the number $n$ of players, a result which was independently discovered by Balcan et al. [2016]. With a more intricate analysis, we then improve this sample complexity to be logarithmic in $n$, but at the cost of a weaker relaxation. Our algorithm, which runs in polynomial time, directly computes a cost allocation that empirically satisfies the core property, i.e., it satisfies the core property on all of the samples. We argue, by leveraging VC-dimension and Rademacher complexity-based generalization bounds, that the same cost allocation will likely satisfy the core property on newly drawn samples as well. We also propose a stronger notion of the approximate core, and prove that it cannot be computed by any algorithm. This hardness result is information theoretic and is not due to running time limitations. The proofs in this section are deferred to Appendix B.

We begin by defining three notions of the approximate core: the probably approximately stable (Balcan et al. [2016]), mostly approximately stable, and probably mostly approximately stable cores.

**Definition 3.** *Given $\delta, \epsilon > 0$, a cost allocation $\psi$ such that $\sum_{i \in N} \psi_i = C(N)$ is in*

- *the **probably approximately stable** core if $\Pr_{S \sim \mathcal{D}} \left[ \sum_{i \in S} \psi_i \leq C(S) \right] \geq 1 - \delta$ for all $\mathcal{D}$ (see Balcan et al. [2015]),*

- *the **mostly approximately stable** core over $\mathcal{D}$ if $(1 - \epsilon) \sum_{i \in S} \psi_i \leq C(S)$ for all $S \subseteq N$,*

- *the **probably mostly approximately stable** core if $\Pr_{S \sim \mathcal{D}} \left[ (1 - \epsilon) \sum_{i \in S} \psi_i \leq C(S) \right] \geq 1 - \delta$ for all $\mathcal{D}$,*

For each of these notions, our goal is to *efficiently compute* a cost allocation in the approximate core, in the following sense.

**Definition 4.** *A cost allocation $\psi$ is efficiently computable for the class of functions $\mathcal{C}$ over distribution $\mathcal{D}$, if for all $C \in \mathcal{C}$ and any $\Delta, \delta, \epsilon > 0$, given $C(N)$ and $m = \text{poly}(n, 1/\Delta, 1/\delta, 1/\epsilon)$ samples $(S_j, C(S_j))$ with each $S_j$ drawn i.i.d. from distribution $\mathcal{D}$, there exists an algorithm that computes $\psi$ with probability at least $1 - \Delta$ over both the samples and the choices of the algorithm.*

We refer to the number of samples required to compute approximate cores as the *sample complexity* of the algorithm. We first present our result for computing a probably approximately stable core with sample complexity that is linear in the number of players, which was also independently discovered by Balcan et al. [2016].

**Theorem 1.** *The class of functions with a non-empty core has cost shares in the probably approximately stable core that are efficiently computable. The sample complexity is*

$$O \left( \frac{n + \log(1/\Delta)}{\delta} \right).$$

The full proof of Theorem 1 is in Appendix B, and can be summarized as follows: We define a class of halfspaces which contains the core. Since we assume that $C$ has a non-empty core, there exists a cost allocation $\psi$ in this class of halfspaces that satisfies both the core property on all the samples and the balance property. Given a set of samples, such a cost allocation can be computed with a simple linear program. We then use the VC-dimension of the class of halfspaces to show that the performance on the samples generalizes well to the performance on the distribution $\mathcal{D}$.

We next show that the sample complexity dependence on $n$ can be improved from linear to logarithmic if we relax the goal from computing a cost allocation in the probably approximately stable core to computing one in the probably mostly approximately stable core instead. The sample complexity of our algorithm also depends on the *spread* of the function $C$, defined as $\frac{\max_S C(S)}{\min_{S \neq \emptyset} C(S)}$ (we assume $\min_{S \neq \emptyset} C(S) > 0$).

**Theorem 2.** *The class of functions with a non-empty core has cost allocations in the probably mostly approximately stable core that are efficiently computable with sample complexity*

$$\left(\frac{1-\epsilon}{\epsilon\delta}\right)^2 \left(128\tau(C)^2 \log(2n) + 8\tau(C)^2 \log(2/\Delta)\right) = O\left(\left(\frac{\tau(C)}{\epsilon\delta}\right)^2 (\log n + \log(1/\Delta))\right).$$

*where $\tau(C) = \frac{\max_S C(S)}{\min_{S \neq \emptyset} C(S)}$ is the spread of $C$.*

The full proof of Theorem 2 is in Appendix B. Its main steps are:

1. We find a cost allocation which satisfies the core property on all samples, restricting the search to cost allocations with bounded $\ell_1$-norm. Such a cost allocation can be found efficiently since the space of such cost allocations is convex.

2. The analysis begins by bounding the $\ell_1$-norm of any vector in the core (Lemma 3). Combined with the assumption that the core is non-empty, this implies that a cost allocation $\psi$ satisfying the previous conditions exists.

3. Let $[x]_+$ denote the function $x \mapsto \max(x, 0)$. Consider the following "loss" function:

$$\left[\frac{\sum_{i \in S} \psi_i}{C(S)} - 1\right]_+$$

This loss function is convenient since it is equal to $0$ if and only if the core property is satisfied for $S$ and it is 1-Lipschitz, which is used in the next step.

4. Next, we bound the difference between the empirical loss and the expected loss for all $\psi$ with a known result using the Rademacher complexity of linear predictors with low $\ell_1$ norm over $\rho$-Lipschitz loss functions (Theorem 10).

5. Finally, given $\psi$ which approximately satisfies the core property in expectation, we show that $\psi$ is in the probably mostly approximately stable core by Markov's inequality (Lemma 4).

Since we obtained a probably mostly approximately stable core, a natural question is if it is possible to compute cost allocations that are mostly approximately stable over natural distributions. The answer is negative in general: even for the restricted class of monotone submodular functions, which always have a solution in the core, the core cannot be mostly approximated from samples, even over the uniform distribution. The full proof of this impossibility theorem is in Appendix B.

**Theorem 3.** *Cost allocations $\psi$ in the $(1/2 + \epsilon)$-mostly approximately stable core, i.e., such that for all $S$,*

$$\left(\frac{1}{2} + \epsilon\right) \cdot \sum_{i \in S} \psi_i \leq C(S),$$

*cannot be computed for monotone submodular functions over the uniform distribution, for any constant $\epsilon > 0$.*

## 4 Approximating the Shapley Value from Samples

We turn our attention to the STATISTICAL COST SHARING problem in the context of the Shapley value. Since the Shapley value exists and is unique for all functions, a natural relaxation is to simply approximate this value from samples. The distributions we consider in this section are the uniform distribution, and more generally product distributions, which are the standard distributions studied in the learning literature for combinatorial functions Balcan and Harvey [2011], Balcan et al. [2012], Feldman and Kothari [2014], Feldman and Vondrak [2014]. It is easy to see that we need some restrictions on the distribution $\mathcal{D}$ (for example, if the empty set if drawn with probability one, the Shapley value cannot be approximated).

For submodular functions with bounded curvature, we prove approximation bounds when samples are drawn from the uniform or a bounded product distribution, and also show that the bound for the uniform distribution is tight. However, we show that the Shapley value cannot be approximated from samples even for coverage functions (which are a special case of submodular functions) and the uniform distribution. Since coverage functions are learnable from samples, this implies the counter-intuitive observation that learnability does not imply that the Shapley value is approximable from samples. We defer the full proofs to Appendix C.

**Definition 5.** *An algorithm $\alpha$-approximates, $\alpha \in (0, 1]$, the Shapley value of cost functions $\mathcal{C}$ over distribution $\mathcal{D}$, if, for all $C \in \mathcal{C}$ and all $\delta > 0$, given $\mathrm{poly}(n, 1/\delta, 1/1-\alpha)$ samples from $\mathcal{D}$, it computes Shapley value estimates $\tilde{\phi}_C$ such that $\alpha\phi_i \leq \tilde{\phi}_i \leq \frac{1}{\alpha}\phi_i$ for all $i \in N$ such that $\phi_i \geq 1/\mathrm{poly}(n)$[1] with probability at least $1 - \delta$ over both the samples and the choices made by the algorithm.*

We consider submodular functions with bounded curvature, a common assumption in the submodular maximization literature Iyer and Bilmes [2013], Iyer et al. [2013], Sviridenko et al. [2015], Vondrák [2010]. Intuitively, the curvature of a submodular function bounds by how much the marginal contribution of an element can decrease. This property is useful since the Shapley value of an element can be written as a weighted sum of its marginal contributions over all sets.

**Definition 6.** *A monotone submodular function $C$ has curvature $\kappa \in [0, 1]$ if $C_{N\setminus\{i\}}(i) \geq (1 - \kappa)C(i)$ for all $i \in N$. This curvature is bounded if $\kappa < 1$.*

An immediate consequence of this definition is that $C_S(i) \geq (1 - \kappa)C_T(i)$ for all $S, T$ such that $i \notin S \cup T$ by monotonicity and submodularity. The main tool used is estimates $\tilde{v}_i$ of expected marginal contributions $v_i = \mathrm{E}_{S \sim \mathcal{D}|i \notin S}[C_S(i)]$ where $\tilde{v}_i = \mathrm{avg}(\mathcal{S}_i) - \mathrm{avg}(\mathcal{S}_{-i})$ is the difference between the average value of samples containing $i$ and the average value of samples not containing $i$.

**Theorem 4.** *Monotone submodular functions with bounded curvature $\kappa$ have Shapley value that is $\sqrt{1 - \kappa} - \epsilon$ approximable from samples over the uniform distribution, which is tight, and $1 - \kappa - \epsilon$ approximable over any bounded product distribution for any constant $\epsilon > 0$.*

Consider the algorithm which computes $\tilde{\phi}_i = \tilde{v}_i$. Note that $\phi_i = \mathrm{E}_{\sigma}[C(A_{\sigma<i} \cup \{i\}) - C(A_{\sigma<i})] \geq (1 - \kappa)v_i > \frac{1-\kappa}{1+\epsilon}\tilde{v}_i > (1 - \kappa - \epsilon)\tilde{v}_i$ where the first inequality is by curvature and the second by Lemma 5 which shows that the estimates $\tilde{v}_i$ of $v_i$ are arbitrarily good. The other direction follows similarly. The $\sqrt{1 - \kappa}$ result is the main technical component of the upper bound. We describe two main steps:

1. The expected marginal contribution $\mathrm{E}_{S \sim \mathcal{U}|i \notin S, |S|=j}[C_S(i)]$ of $i$ to a uniformly random set $S$ of size $j$ is decreasing in $j$, which is by submodularity.

2. Since a uniformly random set has size concentrated close to $n/2$, this implies that roughly half of the terms in the summation $\phi_i = (\sum_{j=0}^{n-1} \mathrm{E}_{S \sim \mathcal{U}_j|i \notin S}[C_S(i)])/n$ are greater than $v_i$ and the other half of the terms are smaller.

For the tight lower bound, we show that there exists two functions that cannot be distinguished from samples w.h.p. and that have an element with Shapley value which differs by an $\alpha^2$ factor.

We show that the Shapley value of coverage (and submodular) functions are not approximable from samples in general, even though coverage functions are PMAC-learnable ( Balcan and Harvey [2011]) from samples over any distribution Badanidiyuru et al. [2012].

**Theorem 5.** *There exists no constant $\alpha > 0$ such that coverage functions have Shapley value that is $\alpha$-approximable from samples over the uniform distribution.*

## 5 Data Dependent Shapley Value

The general impossibility result for computing the Shapley value from samples arises from the fact that the concept was geared towards the query model, where the algorithm can ask for the cost of any set $S \subseteq N$. In this section, we develop an analogue that is distribution-dependent. We denote it by $\phi^{C,\mathcal{D}}$ with respect to both $C$ and $\mathcal{D}$. We define four natural distribution-dependent axioms resembling

the Shapley value axioms, and then prove that our proposed value is the unique solution satisfying them. This value can be approximated arbitrarily well in the statistical model for all functions. The proofs are deferred to Appendix D. We start by stating the four axioms.

**Definition 7.** *The* data-dependent axioms *for cost sharing functions are:*

- **Balance:** $\sum_{i \in N} \phi_i^{\mathcal{D}} = \mathrm{E}_{S \sim \mathcal{D}}[C(S)]$,

- **Symmetry:** *for all $i$ and $j$, if* $\mathrm{Pr}_{S \sim \mathcal{D}}\left[|S \cap \{i,j\}| = 1\right] = 0$ *then* $\phi_i^{\mathcal{D}} = \phi_j^{\mathcal{D}}$,

- **Zero element:** *for all $i$, if* $\mathrm{Pr}_{S \sim \mathcal{D}}\left[i \in S\right] = 0$ *then* $\phi_i^{\mathcal{D}} = 0$,

- **Additivity:** *for all $i$, if $\mathcal{D}_1$, $\mathcal{D}_2$, $\alpha$, and $\beta$ such that $\alpha + \beta = 1$, $\phi_i^{\alpha \mathcal{D}_1 + \beta \mathcal{D}_2} = \alpha \phi_i^{\mathcal{D}_1} + \beta \phi_i^{\mathcal{D}_2}$ where* $\mathrm{Pr}\left[S \sim \alpha \mathcal{D}_1 + \beta \mathcal{D}_2\right] = \alpha \cdot \mathrm{Pr}\left[S \sim \mathcal{D}_1\right] + \beta \cdot \mathrm{Pr}\left[S \sim \mathcal{D}_2\right]$.

The similarity to the original Shapley value axioms is readily apparent. The main distinction is that we expect these to hold with regard to $\mathcal{D}$, which captures the frequency with which different coalitions $S$ occur. Interpreting the axioms one by one, the balance property ensures that the expected cost is always accounted for. The symmetry axiom states that if two elements always occur together, they should have the same share, since they are indistinguishable. If an element is never observed, then it should have zero share. Finally costs should combine in a linear manner according to the distribution.

The *data-dependent Shapley value* is

$$\phi_i^{\mathcal{D}} := \sum_{S \,:\, i \in S} \mathrm{Pr}\left[S \sim \mathcal{D}\right] \cdot \frac{C(S)}{|S|}.$$

Informally, for all set $S$, the cost $C(S)$ is divided equally between all elements in $S$ and is weighted with the probability that $S$ occurs according to $\mathcal{D}$. The main appeal of this cost allocation is the following theorem.

**Theorem 6.** *The data-dependent Shapley value is the unique value satisfying the four data-dependent axioms.*

The data-dependent Shapley value can be approximated from samples with the following empirical data-dependent Shapley value:

$$\tilde{\phi}_i^{\mathcal{D}} = \frac{1}{m} \sum_{S_j \,:\, i \in S_j} \frac{C(S_j)}{|S_j|}.$$

These estimates are arbitrarily good with arbitrarily high probability.

**Theorem 7.** *The empirical data-dependent Shapley value approximates the data-dependent Shapley value arbitrarily well, i.e.,*

$$|\tilde{\phi}_i^{\mathcal{D}} - \phi_i^{\mathcal{D}}| < \epsilon$$

*with* $\mathrm{poly}(n, 1/\epsilon, 1/\delta)$ *samples and with probability at least $1 - \delta$ for any $\delta, \epsilon > 0$.*

## 6 Discussion and Future Work

We follow a recent line of work that studies classical algorithmic problems from a statistical perspective, where the input is restricted to a collection of samples. Our results fall into two categories, we give results for approximating the Shapley value and the core and propose new cost sharing concepts that are tailored for the statistical framework. We use techniques from multiple fields that encompass statistical machine learning, combinatorial optimization, and, of course, cost sharing. The cost sharing literature being very rich, the number of directions for future work are considerable. Obvious avenues include studying other cost sharing methods in this statistical framework, considering other classes of functions to approximate known methods, and improving the sample complexity of previous algorithms. More conceptually, an exciting modeling question arises when designing "desirable" axioms from data. Traditionally these axioms only depended on the cost function, whereas in this model they can depend on both the cost function and the distribution, providing an interesting interplay.

## Footnotes

[1]See Appendix C for general definition.

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
