[Supplementary Material · paper.pdf]

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

# Appendix

## A Concentration Bounds

**Lemma 1** (Chernoff Bound). *Let $X_1, \ldots, X_n$ be independent indicator random variables with values in $\{0, 1\}$. Let $X = \sum_{i=1}^{n} X_i$ and $\mu = \mathrm{E}[X]$. For $0 < \delta < 1$,*

$$\Pr\left[X \leq (1 - \delta)\mu\right] \leq e^{-\mu\delta^2/2} \quad and \quad \Pr\left[X \geq (1 + \delta)\mu\right] \leq e^{-\mu\delta^2/3}.$$

**Lemma 2** (Hoeffding's inequality). *Let $X_1, \ldots, X_n$ be independent random variables with values in $[0, b]$. Let $X = \frac{1}{m}\sum_{i=1}^{m} X_i$ and $\mu = \mathrm{E}[X]$. Then for every $0 < \epsilon < 1$,*

$$\Pr\left[|X - \mathrm{E}[X]| \geq \epsilon\right] \leq 2e^{-2m\epsilon^2/b^2}.$$

## B Missing Definitions and Analysis from Section 3

We first show the result with linear sample complexity, then the result with logarithmic sample complexity, and finally the impossibility result.

### B.1 Linear sample complexity for probably approximately stable core

We first state the generalization error obtained for a class of functions with VC-dimension $d$.

**Theorem 8** (Shalev-Shwartz and Ben-David [2014], Theorem 6.8). *Let $\mathcal{H}$ be a hypothesis class of functions from a domain $\mathcal{X}$ to $\{-1, 1\}$ and $f : \mathcal{X} \mapsto \{-1, 1\}$ be some "correct" function. Assume that $\mathcal{H}$ has VC-dimension $d$. Then, there is an absolute constant $c$ such that with $m \geq c(d + \log(1/\Delta))/\delta$ i.i.d. samples $\mathbf{x}^1, \ldots, \mathbf{x}^m \sim \mathcal{D}$,*

$$\left| \Pr_{\mathbf{x} \sim \mathcal{D}} [h(\mathbf{x}) \neq f(\mathbf{x})] - \frac{1}{m} \sum_{i=1}^{m} \mathbb{1}_{h(\mathbf{x}^i) \neq f(\mathbf{x}^i)} \right| \leq \delta$$

*for all $h \in \mathcal{H}$, with probability $1 - \Delta$ over the samples.*

We use a special case of the class of halfspaces, for which we know the VC-dimension.

**Theorem 9** (Shalev-Shwartz and Ben-David [2014], Theorem 9.2). *The class of functions $\{\mathbf{x} \mapsto \mathrm{sign}(\mathbf{w}^\intercal \mathbf{x}) \ : \ \mathbf{w} \in \mathbb{R}^n\}$ has VC-dimension $n$.*

We first define a class of functions that contains the core, and prove that it has low VC-dimension. Given a sample $S$, define $\mathbf{x}^S$ such that $x_i^S = \mathbb{1}_{i \in S}$ for $i \in [n]$ and $x_{n+1}^S = C(S)$. Note that $\mathrm{sign}\left(\sum_{i=1}^{n} \psi_i x_i^S - x_{n+1}^S\right) = -1$ if the core property is satisfied for sample $S$. We now bound the VC-dimension of this hypothesis class of functions induced by cost allocations $\psi$.

**Corollary 1.** *The class of functions $\mathcal{H}^{core} = \{\mathbf{x} \mapsto sign(\sum_{i=1}^{n} \psi_i x_i - x_{n+1}) \ : \ \psi \in \mathbb{R}^n, \sum_i \psi_i = C(N)\}$ has VC-dimension at most $n + 1$.*

*Proof.* We combine the observation that $\{\mathbf{x} \mapsto \mathrm{sign}(\sum_{i=1}^{n} w_i x_i - x_{n+1}) \ : \ \mathbf{w} \in \mathbb{R}^n, \sum_i w_i = C(N)\} \subseteq \{\mathbf{x} \mapsto \mathrm{sign}(\mathbf{w}^\intercal \mathbf{x}) \ : \ \mathbf{w} \in \mathbb{R}^{n+1}\}$ with the well-known fact that the VC-dimension of $\mathcal{H}'$ is at most the VC-dimension of $\mathcal{H}$ for $\mathcal{H}' \subseteq \mathcal{H}$. $\square$

It remains to show how to optimize over functions in this class.

**Theorem 1.** *The class of functions with a non-empty core has cost shares in the probably approximately stable core that are efficiently computable. The sample complexity is*

$$O\left(\frac{n + \log(1/\Delta)}{\delta}\right).$$

*Proof.* Let $\psi$ be a cost allocation which satisfies both the core property on all the samples and the balance property, i.e., $\sum_{i \in S} \psi_i \leq C(S)$ for all samples $S$ and $\sum_{i \in N} \psi_i = C(N)$. Note that such a

cost allocation exists since we assume that $C$ has a non-empty core. Given the set of samples, it can be computed with a simple linear program. We argue that $\psi$ is probably approximately stable.

Define $h(x) = \text{sign}\left(\sum_{i=1}^{n} \psi_i x_i^S - x_{n+1}^S\right)$ and $f(\mathbf{x}) = -1$ for all $\mathbf{x}$. Since the core property is satisfied on all the samples, $\frac{1}{m}\sum_{i=1}^{m} \mathbb{1}_{h(x)\neq f(x)} = 0$. Thus, by Theorem 8,

$$
\begin{aligned}
\Pr_{S\sim\mathcal{D}}\left[\sum_{i\in S}\psi_i \le C(S)\right] &= 1 - \Pr_{\mathbf{x}^S:S\sim\mathcal{D}}\left[\text{sign}\left(\sum_{i=1}^{n}\psi_i x_i^S - x_{n+1}^S\right) \neq -1\right] \\
&= 1 - \Pr_{\mathbf{x}^S:S\sim\mathcal{D}}\left[h\left(\mathbf{x}^S\right) \neq f\left(\mathbf{x}^S\right)\right] \\
&= 1 - \left| \Pr_{\mathbf{x}^S:S\sim\mathcal{D}}\left[h\left(\mathbf{x}^S\right) \neq f\left(\mathbf{x}^S\right)\right] - \frac{1}{m}\sum_{i=1}^{m}\mathbb{1}_{h(\mathbf{x}^S)\neq f(\mathbf{x}^S)}\right| \\
&\ge 1 - \delta
\end{aligned}
$$

with $O((n + \log(1/\Delta))/\delta)$ samples. $\qquad\square$

## B.2 Logarithmic sample complexity for probably mostly approximately stable cores

The following result follows from the Rademacher complexity of linear classes.

**Theorem 10** (Shalev-Shwartz and Ben-David [2014], Theorem 26.15). *Suppose that $\mathcal{D}$ is a distribution over $\mathcal{X} \times \mathbb{R}$ such that with probability 1 we have that $\|\mathbf{x}\|_\infty \le R$. Let $\mathcal{H} = \{\mathbf{w} \in \mathbb{R}^d : \|\mathbf{w}\|_1 \le B\}$ and let $\ell : \mathcal{H} \times (\mathcal{X} \times \mathbb{R}) \to \mathbb{R}$ be a loss function of the form*

$$\ell(\mathbf{w}, (\mathbf{x}, y)) = \phi(\mathbf{w}^\mathsf{T}\mathbf{x}, y)$$

*such that for all $y \in \mathbb{R}$, $a \mapsto \phi(a, y)$ is an $\rho$-Lipschitz function and such that $\max_{a\in[-BR,BR]}|\phi(a, y)| \le c$. Then, for all $\mathbf{w} \in \mathcal{H}$ and any $\Delta \in (0, 1)$, with probability of at least $1 - \Delta$ over $m$ i.i.d. samples $(\mathbf{x}_1, y_1), \ldots, (\mathbf{x}_m, y_m)$ from $\mathcal{D}$,*

$$\mathop{\mathrm{E}}_{(\mathbf{x},y)\sim\mathcal{D}}[\ell(\mathbf{w}, (\mathbf{x}, y))] \le \frac{1}{m}\sum_{i=1}^{m}\ell(\mathbf{w}, (\mathbf{x}_i, y_i)) + 2\rho BR\sqrt{\frac{2\log(2d)}{m}} + c\sqrt{\frac{2\log(2/\Delta)}{m}}.$$

We first bound the $\ell_1$ norm of vectors in the core to bound the space of linear functions that we search over.

**Lemma 3.** *Assume that $\psi$ is a vector in the core, then $\|\psi\|_1 \le 2\max_S |C(S)|$.*

*Proof.* Fix some vector $\psi$ in the core. Let $A$ be the set of elements $i$ such that $\psi_i \ge 0$ and $B$ be the remaining elements. First note that

$$\sum_{i\in A}\psi_i \le C(A) \le \max_S |C(S)|$$

where the first inequality is by the core property. Next, note that

$$0 \le C(N) = \sum_{i\in A}\psi_i + \sum_{i\in B}\psi_i \le \max_S |C(S)| + \sum_{i\in B}\psi_i$$

where the equality is by the balance property, so $\sum_{i\in B}\psi_i \ge -\max_S |C(S)|$. Thus,

$$\|\psi\|_1 = \sum_{i\in A}\psi_i - \sum_{i\in B}\psi_i \le \max_S |C(S)| + \max_S |C(S)|.$$

$\qquad\square$

We can thus focus on bounded cost allocations $\psi \in \mathcal{H}$ where

$$\mathcal{H} := \left\{\psi \ : \ \psi \in \mathbb{R}^n, \|\psi\|_1 \le 2\max_S |C(S)|\right\}.$$

The next lemma shows that if the core property approximately holds in expectation, then it is likely to approximately hold.

**Lemma 4.** *For any $0 < \epsilon, \delta < 1$ and cost allocation $\psi$,*

$$\operatorname*{E}_{S \sim \mathcal{D}}\left[\left[\frac{\sum_{i \in S} \psi_i}{C(S)} - 1\right]_+\right] \leq \frac{\epsilon \delta}{1 - \epsilon} \Rightarrow \operatorname*{Pr}_{S \sim \mathcal{D}}\left[(1 - \epsilon)\sum_{i \in S} \psi_i \leq C(S)\right] \geq 1 - \delta.$$

*Proof.* For any $a > 0$ and nonnegative random variable $X$, by Markov's inequality we have $\Pr[X \leq a] \geq 1 - E[X]/a$. By letting $a = \epsilon/(1 - \epsilon)$, $X = \left[(\sum_{i \in S} \psi_i)/C(S) - 1\right]_+$, and observing that

$$\left[\frac{\sum_{i \in S} \psi_i}{C(S)} - 1\right]_+ \leq \frac{\epsilon}{1 - \epsilon} \Rightarrow \frac{\sum_{i \in S} \psi_i}{C(S)} - 1 \leq \frac{\epsilon}{1 - \epsilon} \Rightarrow (1 - \epsilon)\sum_{i \in S} \psi_i \leq C(S),$$

we obtain $\Pr_{S \sim \mathcal{D}}\left[(1 - \epsilon)\sum_{i \in S} \psi_i \leq C(S)\right] \geq 1 - \delta$. $\qquad\square$

Combining Theorem 10, Lemma 3, and Lemma 4, we obtain the main result.

**Theorem 2.** *The class of functions with a non-empty core has cost allocations in the probably mostly approximately stable core that are efficiently computable with sample complexity*

$$\left(\frac{1 - \epsilon}{\epsilon \delta}\right)^2 \left(128\tau(C)^2 \log(2n) + 8\tau(C)^2 \log(2/\Delta)\right) = O\left(\left(\frac{\tau(C)}{\epsilon \delta}\right)^2 (\log n + \log(1/\Delta))\right).$$

*where $\tau(C) = \frac{\max_S C(S)}{\min_{S \neq \emptyset} C(S)}$ is the spread of $C$.*

*Proof.* Fix $C \in \mathcal{C}$. Suppose we are given $m$ samples from $\mathcal{D}$. We pick $\psi^\star \in \mathcal{H}$ such that core property holds on all the samples and such that the balance property holds ($\sum_{i \in N} \psi_i = C(N)$). This cost allocation $\psi^\star$ can be found efficiently since the collection of such $\psi$ is a convex set. By the assumption that $C$ has at least one vector in the core and by Lemma 3, such a $\psi^\star$ exists. Given $S \sim \mathcal{D}$, define $\mathbf{x}^S$ such that $x_i^S = \mathbb{1}_{i \in S}/C(S)$. Fix $y = 1$. Define the loss function $\ell$ as follows,

$$\ell\left(\psi, \left(\mathbf{x}^S, y\right)\right) := \left[\psi^\intercal \mathbf{x}^S - y\right]_+ = \left[\frac{\sum_{i \in S} \psi_i}{C(S)} - 1\right]_+$$

We wish to use Theorem 10 with $R = 1/\min_{S \neq \emptyset} |C(S)|$, $B = 2\max_S |C(S)|$, $\phi(a, y) = [a - y]_+$, $\rho = 1$, and $c = \tau(C)$. We verify that all the conditions hold. First note that without loss of generality, samples where $S = \emptyset$ can be ignored, so $\left\|\mathbf{x}^S\right\|_\infty \leq 1/\min_{S \neq \emptyset} |C(S)|$ for all $S$. Next, $\|\psi\|_1 \leq 2\max_S |C(S)|$ for $\psi \in \mathcal{H}$ by definition of $\mathcal{H}$. The loss function $\ell$ is of the form $\ell(\psi, (\mathbf{x}, y)) = \phi(\psi^\intercal \mathbf{x}, y) = [\psi^\intercal \mathbf{x} - y]_+$ such that $a \mapsto \phi(a, y) = [a - y]_+$ is an 1-Lipschitz function and such that $\max_{a \in [-BR, BR]} |\phi(a, y)| \leq 2\max_S |C(S)|/\min_{S \neq \emptyset} |C(S)| = 2\tau(C)$. In addition, note that

$$\frac{1}{m} \sum_{i=1}^m \ell\left(\psi^\star, \left(\mathbf{x}^{S_i}, 1\right)\right) = 0$$

since the core property holds on all the samples. Thus, by Theorem 10,

$$\operatorname*{E}_{S \sim \mathcal{D}}\left[\frac{\sum_{i \in S} \psi_i^\star}{C(S)} - 1\right]_+ = \operatorname*{E}_{\mathbf{x}^S : S \sim \mathcal{D}}\left[\ell\left(\psi^\star, \left(\mathbf{x}^S, 1\right)\right)\right] \leq 4\tau(C)\sqrt{\frac{2\log(2n)}{m}} + \tau(C)\sqrt{\frac{2\log(2/\Delta)}{m}}.$$

Choose any $\epsilon, \delta > 0$. If the number of samples $m$ is chosen as in the statement of the theorem, then the righthand side of the above inequality will be less than $\frac{\epsilon \delta}{1 - \epsilon}$. Thus by Lemma 4,

$$\operatorname*{Pr}_{S \sim \mathcal{D}}\left[(1 - \epsilon)\sum_{i \in S} \psi_i \leq C(S)\right] \geq 1 - \delta,$$

which completes the proof. $\qquad\square$

### B.3 Hardness of mostly approximately stable core

We now give the missing analysis for the impossibility result of approximating the core.

**Theorem 3.** *Cost allocations $\psi$ in the $(1/2 + \epsilon)$-mostly approximately stable core, i.e., such that for all $S$,*

$$\left(\frac{1}{2} + \epsilon\right) \cdot \sum_{i \in S} \psi_i \leq C(S),$$

*cannot be computed for monotone submodular functions over the uniform distribution, for any constant $\epsilon > 0$.*

*Proof.* The ground set of elements is partitioned in $\epsilon^{-1}$ sets $A_1, \ldots A_{\epsilon^{-1}}$ of size $\epsilon n$ for some small constant $\epsilon > 0$. Let $\mathcal{C} = \{C^{A_i} : i \in [\epsilon^{-1}]\}$ where

$$C^A(S) = |(N \setminus A) \cap S| + \min\left(|A \cap S|, (1 + \epsilon)\frac{\epsilon n}{2}\right).$$

The expected number of elements of $A_i$ in a sample $S$ from the uniform distribution is $|A_i|/2 = \epsilon n/2$, so by the Chernoff bound

$$\Pr\left[|A_i \cap S| \geq (1 + \epsilon)\frac{n\epsilon}{2}\right] \leq e^{-\frac{\epsilon^2 n}{6}},$$

Thus, by a union bound, $C^{A_i}(S) = |S|$ over all $i$ and all samples $S$ with probability $1 - O(e^{-n})$ and we henceforth assume this is the case. It is therefore impossible to learn any information about the partition $A_1, \ldots A_{\epsilon^{-1}}$ from samples. Any cost allocation $\psi$ computed by an algorithm given samples from $C^{A_i}$ is thus *independent* of $i$.

Next, consider such a cost allocation $\psi$ independent of $i$ satisfying the balance property. There exists $A_i$ such that $\sum_{j \in A_i} \psi_j > (1 - \epsilon)n\epsilon$ since $\sum_{j \in N} \psi_j = C(N) > (1 - \epsilon)n$ by the balance property. In addition, $C^{A_i}(A_i) = (1 + \epsilon)\epsilon n/2$. We obtain

$$\sum_{j \in A_i} \psi_j > (1 - \epsilon)n\epsilon = \frac{(1 - \epsilon)}{(1 + \epsilon)}2C^{A_i}(A_i).$$

Thus, the core property is violated by a $1/2 + \epsilon'$ factor for set $A_i$ and function $C^{A_i}$, and for any constant $\epsilon' > 0$ by picking $\epsilon$ sufficiently small. □

## C  Missing Analysis from Section 4

We give a complete definition of approximate Shapley values.

**Definition 8.** *An algorithm $\alpha$-approximates, $\alpha \in (0, 1]$, the Shapley value of a family of cost functions $\mathcal{C}$ over distribution $\mathcal{D}$, if, for all $C \in \mathcal{C}$ and all $\delta > 0$, given $\mathrm{poly}(n, 1/\delta, 1/1-\alpha)$ samples from $\mathcal{D}$, it computes Shapley value estimates $\tilde{\phi}_C$ such that for*

- *positive bounded Shapley value, if $\phi_i \geq 1/\mathrm{poly}(n)$, then $\alpha\phi_i \leq \tilde{\phi}_i \leq \frac{1}{\alpha}\phi_i$;*

- *negative bounded Shapley value, if $\phi_i \leq -1/\mathrm{poly}(n)$, then $\frac{1}{\alpha}\phi_i \leq \tilde{\phi}_i \leq \alpha\phi_i$;*

- *small Shapley value, if $|\phi_i| < 1/\mathrm{poly}(n)$, then $|\phi_i - \tilde{\phi}_i| = o(1)$.*

*for all $i \in N$ with probability at least $1 - \delta$ over both the samples and the choices made by the algorithm.*

Next, we show that the estimates $\tilde{v}_i$ of the expected marginal contribution $v_i = \mathrm{E}_{S \sim \mathcal{D}|i \notin S}[C_S(i)]$ are arbitrarily good. Recall that $\mathcal{S}_i$ and $\mathcal{S}_{-i}$ are the collections of samples containing element $i$ and not containing it respectively and that $\mathrm{avg}(\mathcal{S}) = (\sum_{S \in \mathcal{S}} C(S))/|\mathcal{S}|$ is the average value of the samples in $\mathcal{S}$. Let $v_i = C(i)$ and $\tilde{v}_i = \mathrm{avg}(\mathcal{S}_i) - \mathrm{avg}(\mathcal{S}_{-i})$.

**Lemma 5.** *The expected marginal contribution of an element $i$ to a random set from a bounded product distribution $\mathcal{D}$ not containing $i$ is estimated arbitrarily well by $\tilde{v}_i$, i.e., for all $i \in N$ and given $\mathrm{poly}(n, 1/\delta, 1/\epsilon)$ samples,*

$$
\begin{aligned}
(1-\epsilon)v_i \leq \tilde{v}_i \leq (1+\epsilon)v_i && \text{if } v_i \geq 1/\mathrm{poly}(n) \\
|v_i - \tilde{v}_i| \leq \epsilon && \text{if } |v_i| < 1/\mathrm{poly}(n) \\
(1+\epsilon)v_i \leq \tilde{v}_i \leq (1-\epsilon)v_i && \text{if } v_i \leq -1/\mathrm{poly}(n)
\end{aligned}
$$

*with probability at least $1 - \delta$ for any $\delta > 0$.*

*Proof.* Note that

$$
v_i = \mathop{\mathrm{E}}_{S \sim \mathcal{D} | i \notin S}[C_S(i)] = \mathop{\mathrm{E}}_{S \sim \mathcal{D} | i \notin S}[C(S \cup i)] - \mathop{\mathrm{E}}_{S \sim \mathcal{D} | i \notin S}[C(S)] = \mathop{\mathrm{E}}_{S \sim \mathcal{D} | i \in S}[C(S)] - \mathop{\mathrm{E}}_{S \sim \mathcal{D} | i \notin S}[C(S)].
$$

where the second equality is since $\mathcal{D}$ is a product distribution. We also have that $\mathrm{E}[\mathrm{avg}(\mathcal{S}_i)] = \mathrm{E}_{S \sim \mathcal{D} | i \in S}[C(S)]$ and $\mathrm{E}[\mathrm{avg}(\mathcal{S}_{-i})] = \mathrm{E}_{S \sim \mathcal{D} | i \notin S}[C(S)]$. Since marginal probabilities of the product distributions are assumed to be bounded from below and above by $1/\mathrm{poly}(n)$ and $1 - 1/\mathrm{poly}(n)$ respectively, $|\mathcal{S}_i| = m/\mathrm{poly}(n)$ and $|\mathcal{S}_{-i}| = m/\mathrm{poly}(n)$ for all $i$ by Chernoff bound. In addition, $\max_S C(S)$ is assumed to be bounded by $\mathrm{poly}(n)$. So by Hoeffding's inequality,

$$
\mathrm{Pr}\left( \left| \mathrm{avg}(\mathcal{S}_i) - \mathop{\mathrm{E}}_{S \sim \mathcal{D} | i \in S}[C(S)] \right| \geq |v_i|\epsilon/2 \right) \leq 2e^{-\frac{m(|v_i|\epsilon)^2}{\mathrm{poly}(n)}},
$$

for $0 < \epsilon < 2/v_i$ and

$$
\mathrm{Pr}\left( \left| \mathrm{avg}(\mathcal{S}_{-i}) - \mathop{\mathrm{E}}_{S \sim \mathcal{D} | i \notin S}[C(S)] \right| \geq |v_i|\epsilon/2 \right) \leq 2e^{-\frac{m(|v_i|\epsilon)^2}{\mathrm{poly}(n)}}.
$$

Thus,

$$
\mathrm{Pr}(|\tilde{v}_i - v_i| \geq |v_i|\epsilon) \leq 2e^{-\frac{m(|v_i|\epsilon)^2}{\mathrm{poly}(n)}}
$$

and, either $(1-\epsilon)v_i \leq \tilde{v}_i \leq (1+\epsilon)v_i$ if $v_i > 0$ or $(1+\epsilon)v_i \leq \tilde{v}_i \leq (1-\epsilon)v_i$ if $v_i < 0$, with probability at least $1 - 2e^{-\frac{m(|v_i|\epsilon)^2}{\mathrm{poly}(n)}}$. If $|v_i| < 1/\mathrm{poly}(n)$, we obtain $|v_i - \tilde{v}_i| < \epsilon$ with a similar analysis without any assumption on $v_i$. Otherwise, the bounds on the estimation hold with probability at least $1 - 2e^{\frac{m\epsilon^2}{\mathrm{poly}(n)}}$. $\qquad\square$

**Theorem 4.** *Monotone submodular functions with bounded curvature $\kappa$ have Shapley value that is $\sqrt{1 - \kappa} - \epsilon$ approximable from samples over the uniform distribution, which is tight, and $1 - \kappa - \epsilon$ approximable over any bounded product distribution for any constant $\epsilon > 0$.*

It remains to improve the bound from $1 - \kappa - \epsilon$ to $\sqrt{1 - \kappa} - \epsilon$ for the uniform distribution. Let

$$
\tilde{\phi}_i = \frac{2 - \kappa}{2\sqrt{1 - \kappa}} \cdot \tilde{v}_i
$$

be the estimated Shapley value. Denote by $\mathcal{U}_j$ the uniform distribution over all sets of size $j$, so

$$
\phi_i = \mathop{\mathrm{E}}_{\sigma}[C_{A_{\sigma < i}}(i)] = \frac{1}{n} \sum_{j=0}^{n-1} \mathop{\mathrm{E}}_{S \sim \mathcal{U}_j | i \notin S}[C_S(i)].
$$

The main idea to improve the loss from $1 - \kappa$ to $\sqrt{1 - \kappa}$ is to observe that $v_i$ can be a factor $1 - \kappa$ away from the contribution $\mathrm{E}_{S \sim \mathcal{U}_{j_l} | i \notin S}[C_S(i)]$ of $j$ to sets of low sizes $j_l \leq L := (1 - \epsilon') \cdot n/2$ or $1 - \kappa$ away from its contribution $\mathrm{E}_{S \sim \mathcal{U}_{j_h} | i \notin S}[C_S(i)]$ to sets of high sizes $j_h \geq H := (1 + \epsilon') \cdot n/2$, but not both, otherwise the curvature property would be violated as illustrated in Figure 1. The following lemma shows that $\mathrm{E}_{S \sim \mathcal{U}_j | i \notin S}[C_S(i)]$ is decreasing in $j$ by submodularity.

**Lemma 6.** *Let $C$ be a submodular function, then for all $j \in \{0, \ldots, n-1\}$ and all $i \in N$,*

$$
\mathop{\mathrm{E}}_{S \sim \mathcal{U}_j | i \notin S}[C_S(i)] \geq \mathop{\mathrm{E}}_{S \sim \mathcal{U}_{j+1} | i \notin S}[C_S(i)]
$$

Figure 1: The expected marginal contribution $E_{S\sim\mathcal{U}_j|i\notin S}[C_S(i)]$ of an element $i$ to a set of size $j$ as a function of $j$. The curvature property implies that any two points are at most a factor $1-\kappa$ from each other. Lemma 6 shows that it is decreasing. Lemma 7 shows that the expected marginal contribution $v_i$ of $i$ to a uniformly random set is approximately between $E_{S\sim\mathcal{U}_L|i\notin S}[C_S(i)]$ and $E_{S\sim\mathcal{U}_H|i\notin S}[C_S(i)]$. The Shapley value of $i$ is the average value of this expected marginal contribution over all integers $j \in \{0, \dots, n-1\}$.

*Proof.* By submodularity,

$$\sum_{S:|S|=j,i\notin S} C_S(i) \geq \sum_{S:|S|=j,i\notin S} \frac{1}{n-j-1} \sum_{i'\notin S\cup\{i\}} C_{S\cup\{i'\}}(i).$$

In addition, observe that by counting in two ways,

$$\sum_{S:|S|=j,i\notin S} \sum_{i'\notin S\cup\{i\}} C_{S\cup\{i'\}}(i) = (j+1) \sum_{S:|S|=j+1,i\notin S} C_S(i).$$

By combining the two previous observations,

$$
\begin{aligned}
\operatorname*{E}_{S\sim\mathcal{U}_j|i\notin S}[C_S(i)] &= \frac{1}{|\{S : |S| = j, i \notin S\}|} \sum_{S:|S|=j,i\notin S} C_S(i) \\
&\geq \frac{1}{\binom{n-1}{j}} \frac{j+1}{n-j-1} \sum_{S:|S|=j+1,i\notin S} C_S(i) \\
&= \frac{1}{|\{S : |S| = j+1, i \notin S\}|} \sum_{S:|S|=j+1,i\notin S} C_S(i) \\
&= \operatorname*{E}_{S\sim\mathcal{U}_{j+1}|i\notin S}[C_S(i)]
\end{aligned}
$$

$\square$

The next lemma shows that for $j$ slightly lower than $n/2$, the expected marginal contribution $v_i$ of element $i$ to a random set cannot be much larger than $\operatorname{E}_{S\sim\mathcal{U}_j|i\notin S}[C_S(i)]$, and similarly for $j$ slightly larger than $n/2$, it cannot be much smaller.

**Lemma 7.** *Let $C$ be a submodular function, then for all $i \in N$,*

$$\left(1 + \frac{e^{-\frac{\epsilon n}{6}}}{1-\kappa}\right) \cdot \operatorname*{E}_{S\sim\mathcal{U}_L|i\notin S}[C_S(i)] \geq v_i \geq \left(1 - e^{-\frac{\epsilon n}{6}}\right) \cdot \operatorname*{E}_{S\sim\mathcal{U}_H|i\notin S}[C_S(i)].$$

*Proof.* By Chernoff bound, $L \leq |S|$ and $|S| \leq H$ with probability at least $1 - e^{-\frac{\epsilon n}{6}}$ each for $S$ drawn from the uniform distribution. Denote the uniform distribution over all sets by $\mathcal{U}$. So,

$$
\begin{aligned}
v_i &= \sum_{j=0}^{n-1} \Pr_{S \sim \mathcal{U}|i \notin S}[|S| = j] \cdot \mathop{\mathrm{E}}_{S \sim \mathcal{U}_j|i \notin S}[C_S(i)] \\
&\geq \sum_{j=0}^{H} \Pr_{S \sim \mathcal{U}|i \notin S}[|S| = j] \cdot \mathop{\mathrm{E}}_{S \sim \mathcal{U}_j|i \notin S}[C_S(i)] \\
&\geq \Pr_{S \sim \mathcal{U}|i \notin S}[|S| \leq H] \cdot \mathop{\mathrm{E}}_{S \sim \mathcal{U}_H|i \notin S}[C_S(i)] && \text{Lemma 6} \\
&\geq (1 - e^{-\frac{\epsilon n}{6}}) \cdot \mathop{\mathrm{E}}_{S \sim \mathcal{U}_H|i \notin S}[C_S(i)].
\end{aligned}
$$

Similarly,

$$
\begin{aligned}
v_i &= \sum_{j=0}^{n-1} \Pr_{S \sim \mathcal{U}|i \notin S}[|S| = j] \cdot \mathop{\mathrm{E}}_{S \sim \mathcal{U}_j|i \notin S}[C_S(i)] \\
&\leq \Pr_{S \sim \mathcal{U}|i \notin S}[|S| < L] \cdot C(i) + \Pr_{S \sim \mathcal{U}|i \notin S}[|S| \geq L] \cdot \mathop{\mathrm{E}}_{S \sim \mathcal{U}_L|i \notin S}[C_S(i)] && \text{Lemma 6} \\
&\leq e^{-\frac{\epsilon n}{6}} \cdot \frac{1}{1-\kappa} \cdot \mathop{\mathrm{E}}_{S \sim \mathcal{U}_L|i \notin S}[C_S(i)] + \mathop{\mathrm{E}}_{S \sim \mathcal{U}_L|i \notin S}[C_S(i)] && \text{curvature}
\end{aligned}
$$

$\square$

We are now ready to prove Theorem 4:

$$
\begin{aligned}
\phi_i &= \frac{1}{n} \sum_{j=0}^{n-1} \mathop{\mathrm{E}}_{S \sim \mathcal{U}_j|i \notin S}[C_S(i)] \\
&= \frac{1}{n} \left( \sum_{j=0}^{H-1} \mathop{\mathrm{E}}_{S \sim \mathcal{U}_j|i \notin S}[C_S(i)] + \sum_{i=H}^{n-1} \mathop{\mathrm{E}}_{S \sim \mathcal{U}_j|i \notin S}[C_S(i)] \right) \\
&\leq \frac{1+\epsilon'}{2} \cdot C(i) + \frac{1}{2} \cdot \mathop{\mathrm{E}}_{S \sim \mathcal{U}_H|i \notin S}[C_S(i)] && \text{Lemma 6} \\
&\leq \frac{1+\epsilon'}{2(1-\kappa)} \cdot v_i + \frac{1}{2(1-e^{-\frac{\epsilon' n}{6}})} \cdot v_i && \text{curvature and Lemma 7} \\
&\leq \left( \frac{2-\kappa}{2(1-\kappa)} + c_1 \epsilon' \right) \cdot v_i \\
&\leq \left( \frac{2-\kappa}{2(1-\kappa)} + c_2 \epsilon' \right) \cdot \tilde{v}_i && \text{Lemma 5} \\
&= \left( \frac{1}{\sqrt{1-\kappa} - c_3 \epsilon'} \right) \cdot \tilde{\phi}_i && \text{definition of } \tilde{\phi}_i
\end{aligned}
$$

for some constants $c_1, c_2, c_3$ and let $\epsilon' = \epsilon/c_3$ to obtain the desired result for any $\epsilon$. Similarly,

$$\phi_i = \frac{1}{n} \sum_{j=0}^{L} \underset{S \sim \mathcal{U}_j | i \notin S}{\mathrm{E}}[C_S(i)] + \frac{1}{n} \sum_{j=L+1}^{n-1} \underset{S \sim \mathcal{U}_j | i \notin S}{\mathrm{E}}[C_S(i)]$$

$$\geq \frac{1-\epsilon'}{2} \cdot \underset{S \sim \mathcal{U}_L | i \notin S}{\mathrm{E}}[C_S(i)] + \frac{1}{2}(C(N) - C(N \setminus \{i\})) \qquad\qquad \text{Lemma 6}$$

$$\geq \frac{1-\epsilon'}{2 \left(1 + \frac{e^{-\frac{\epsilon' n}{6}}}{1-\kappa}\right)} \cdot v_i + \frac{1-\kappa}{2} \cdot v_i \qquad\qquad \text{Lemma 7 and curvature}$$

$$\geq \left(\frac{2-\kappa}{2} - c_1\epsilon'\right) v_i$$

$$\geq \left(\frac{2-\kappa}{2} - c_2\epsilon'\right) \tilde{v}_i \qquad\qquad \text{Lemma 5}$$

$$= \left(\sqrt{1-\kappa} - c_3\epsilon'\right) \tilde{\phi}_i \qquad\qquad \text{definition of } \tilde{\phi}_i$$

$\square$

We show that this approximation is optimal. We begin with a general lemma to derive information theoretic inapproximability results for the Shapley value. This lemma shows that if there exists two functions in $\mathcal{C}$ that cannot be distinguished from samples with high probability and that have an element with Shapley value which differs by an $\alpha^2$ factor, then $\mathcal{C}$ does not have a Shapley value that is $\alpha$-approximable from samples.

**Lemma 8.** *Consider a family of cost functions $\mathcal{C}$, a constant $\alpha \in (0, 1)$, and assume there exist $C^1, C^2 \in \mathcal{C}$ such that:*

- ***Indistinguishable from samples.** With probability $1 - O(e^{-\beta n})$ over $S \sim \mathcal{D}$ for some constant $\beta > 0$,*

$$C^1(S) = C^2(S).$$

- ***Gap in Shapley value.** There exists $i \in N$ such that*

$$\phi_i^{C^1} < \alpha^2 \phi_i^{C^2}.$$

*Then, $\mathcal{C}$ does not have Shapley value that is $\alpha$-approximable from samples over $\mathcal{D}$.*

*Proof.* By a union bound, given $m$ sets $S_1, \ldots, S_m$ drawn i.i.d. from $\mathcal{D}$ with $m$ polynomial in $n$, $C^1(S_j) = C^2(S_j)$ for all $S_j$ with probability $1 - O(e^{-\beta n})$.

Let $C = C^1$ or $C = C^2$ with probability $1/2$ each. Assume the algorithm is given $m$ samples such that $C^1(S_j) = C^2(S_j)$ and consider its (possibly randomized) choice $\tilde{\phi}_i$. Note that $\tilde{\phi}_i$ is independent of the randomization of $C$ since $C^1$ and $C^2$ are indistinguishable to the algorithm. Since $\phi_i^{C^1}/\phi_i^{C^2} < \alpha^2$, $\tilde{\phi}_i$ is at least a factor $\alpha$ away from $\phi_i^C$ with probability at least $1/2$ over the choices of the algorithm and $C$. Label the cost functions so that $\tilde{\phi}_i$ is at least a factor $\alpha$ away from $\phi_i^{C^1}$ with probability at least $1/2$ over the choices of the algorithm. Thus, with $\delta = 1/4$, there exists no algorithm such that for all $C' \in \{C_1, C_2\}$, $\alpha \cdot \phi_i^{C'} \leq \tilde{\phi}_i^{C'} \leq \frac{1}{\alpha} \cdot \phi_i^{C'}$ with probability at least $3/4$ over both the samples and the choices of the algorithm. $\square$

We obtain the inapproximability result by constructing two such functions.

**Theorem 11.** *For every $\kappa < 1$, there exists a hypothesis class of submodular functions with curvature $\kappa$ that have Shapley value that is not $\sqrt{1-\kappa} + \epsilon$-approximable from samples over the uniform distribution, for every constant $\epsilon > 0$.*

*Proof.* We show that an element $i^\star$ can have Shapley value that differs by a factor $1 - \kappa$ for two functions $C^1$ and $C^2$ with curvature $\kappa$ that are indistinguishable from samples. Combining with Lemma 8, we then obtain the negative result. These two functions have a simpler definition via their

Figure 2: The marginal contributions $C_S^1(i^\star)$, (a), and $C_S^2(i^\star)$, (b), of $i^\star$ to a set $S$ of size $j$.

marginal contributions, so we start by defining them in terms of these marginal contributions and we later give their formal definition to show that they are well-defined. The marginal contributions of $i^\star$ are illustrated in Figure 2.

$$C_S^1(i) = \begin{cases} 1 & \text{if } |S| < L \\ 1 - \kappa & \text{otherwise} \end{cases}$$

$$C_S^2(i^\star) = \begin{cases} 1 - \kappa & \text{if } |S| \leq H \\ 1 - \kappa - (|S| - H) \cdot \frac{1 - \kappa - (1-\kappa)^2}{\sqrt{n}} & \text{if } H < |S| \leq H + \sqrt{n} \\ (1 - \kappa)^2 & \text{otherwise} \end{cases}$$

For $i \neq i^\star$:

$$C_S^2(i) = \begin{cases} \frac{L - (1-\kappa)}{L-1} & \text{if } |S| < L - 1 \text{ or } (|S| = L - 1 \text{ and } i^\star \in S) \\ 1 - \kappa & \text{if } (|S| = L - 1 \text{ and } i^\star \notin S) \text{ or } L \leq |S| \leq H \text{ or} \\ & \quad (H \leq |S| \leq H + \sqrt{n} \text{ and } i^\star \notin S) \\ 1 - \kappa - \frac{1 - \kappa - (1-\kappa)^2}{\sqrt{n}} & \text{otherwise} \end{cases}$$

The formal definitions of the functions are

$$C^1(S) = \begin{cases} |S| & \text{if } |S| < L \\ L + (|S| - L) \cdot (1 - \kappa) & \text{otherwise} \end{cases}$$

and

$$C^2(S) = \begin{cases} 1_{i^\star \in S} \cdot (1 - \kappa) + (|S| - 1_{i^\star \in S}) \cdot \frac{L - (1-\kappa)}{L-1} & \text{if } |S| < L \\ L + (|S| - L) \cdot (1 - \kappa) & \text{if } L \leq |S| \leq H \\ & \quad \text{or } (H < |S| \leq H + \sqrt{n} \\ & \quad \text{and } i^\star \notin S) \\ L + (|S| - L) \cdot (1 - \kappa) & \\ \quad + 1 - \kappa - (|S| - H) \cdot \frac{1 - \kappa - (1-\kappa)^2}{\sqrt{n}} & \text{if } H < |S| \leq H + \sqrt{n} \\ & \quad \text{and } i^\star \in S \\ L + (H + \sqrt{n} - L) \cdot (1 - \kappa) + 1_{i^\star \in S} \cdot (1 - \kappa)^2 & \\ \quad + (|S| - 1_{i^\star \in S} - (H + \sqrt{n}))(1 - \kappa - \frac{1 - \kappa - (1-\kappa)^2}{\sqrt{n}}) & \text{otherwise} \end{cases}$$

The Shapley value of $i^\star$ with respect to $C^1$ and $C^2$ is then:

$$\phi_{i^\star}^{C^1} = 1 \cdot \frac{1 - \epsilon'}{2} + (1 - \kappa) \cdot \frac{1 + \epsilon'}{2} \geq \frac{2 - \kappa}{2} - \epsilon$$

and

$$\phi_{i^\star}^{C^2} \leq (1 - \kappa) \cdot \frac{1 + \epsilon'}{2} + (1 - \kappa)^2 \cdot \frac{1 - \epsilon'}{2} \leq \frac{(1 - \kappa)(2 - \kappa)}{2} + \epsilon$$

for an appropriate choice of $\epsilon'$. Next, by Chernoff bound and a union bound, $L \leq |S| \leq H$ for polynomially many samples $S$ from the uniform distirbution, with probability $1 - e^{-\Omega(n)}$. Thus, $C^1(S) = C^2(S)$ for all samples $S$ with probability $1 - e^{-\Omega(n)}$.

It remains to show that $C^1$ and $C^2$ are submodular with curvature $\kappa$, i.e., for any $S \subseteq T$ and $i \notin T$,

$$C_S(i) \geq C_T(i) \geq (1 - \kappa)C_S(i),$$

which is immediate for $C^1$. Regarding $C^2$, it is also immediate that $C_S^2(i^\star) \geq C_T^2(i^\star) \geq (1 - \kappa)C_S^2(i^\star)$. For $i \neq i^\star$, observe that

$$\frac{L - (1 - \kappa)}{L - 1} \leq 1 + \epsilon \quad \text{and} \quad 1 - \kappa - \frac{1 - \kappa - (1 - \kappa)^2}{\sqrt{n}} \geq 1 - \kappa - \epsilon,$$

so $C_S^2(i) \geq C_T^2(i) \geq (1 - \kappa - \epsilon)C_S^2(i)$. $\qquad\square$

**Theorem 5.** *There exists no constant $\alpha > 0$ such that coverage functions have Shapley value that is $\alpha$-approximable from samples over the uniform distribution.*

*Proof.* Partition $N$ into two parts $A$ and $B$ of equal size. Consider the following two functions:

$$C^1(S) = \begin{cases} 0 & \text{if } S = \emptyset \\ 1 & \text{if } |S \cap A| > 0, |S \cap B| = 0 \\ \frac{1}{\alpha^2} & \text{if } |S \cap A| = 0, |S \cap B| > 0 \\ 1 + \frac{1}{\alpha^2} & \text{otherwise} \end{cases}$$

$$C^2(S) = \begin{cases} 0 & \text{if } S = \emptyset \\ 1 & \text{if } |S \cap B| > 0, |S \cap A| = 0 \\ \frac{1}{\alpha^2} & \text{if } |S \cap B| = 0, |S \cap A| > 0 \\ 1 + \frac{1}{\alpha^2} & \text{otherwise} \end{cases}$$

These functions are coverage functions with $U = \{a, b_1, \ldots, b_{1/\alpha^2}\}$ and $T_i = \{a\}$ or $T_i = \{b_1, \ldots, b_{1/\alpha^2}\}$. By the Chernoff bound (Lemma 1) with $\delta = 1/2$ and $\mu = n/2$, if $S$ is a sample from the uniform distribution, then

$$\Pr\left[|S \cap A| = 0\right] = \Pr\left[|S \cap B| = 0\right] < \Pr\left[|S \cap B| \leq n/4\right] \leq e^{-n/16},$$

so $C^1(S) \neq C^2(S)$ with probability at most $2e^{-n/16}$. It is easy to see that for any $i$, its Shapley value is either $2/n$ or $2/(\alpha^2 n)$ depending on which partition it is in. Combining this with Lemma 8 concludes the proof. $\qquad\square$

## D   Missing Analysis from Section 5

**Theorem 6.** *The data-dependent Shapley value is the unique value satisfying the four data-dependent axioms.*

We first show that if there exists a value satisfying the axioms, it must be the data-dependent Shapley value. Then, we show that the data-dependent Shapley value satisfies the axioms, which concludes the proof.

**Lemma 9.** *If there exists a value satisfying the four data-dependent Shapley axioms, then this value is the data-dependent Shapley value.*

*Proof.* Define $\mathcal{D}_S$ to be the distribution such that $\Pr\left[S \sim \mathcal{D}_S\right] = 1$. Observe that the unique value satisfying the balance, symmetry, and zero element axioms must satisfy

$$\phi_i^{\mathcal{D}_S} = \begin{cases} \frac{C(S)}{|S|} & \text{if } i \in S \\ 0 & \text{otherwise.} \end{cases}$$

Since $\mathcal{D} = \sum_S \Pr\left[S \sim \mathcal{D}\right] \cdot \mathcal{D}_S$, the unique value satisfying the four axioms must satisfy $\phi_i^{\mathcal{D}} = \sum_S \Pr\left[S \sim \mathcal{D}\right] \cdot \phi_i^{\mathcal{D}_S} = \sum_{S:i \in S} \Pr\left[S \sim \mathcal{D}\right] \cdot C(S)/|S|$ where the first equality is by additivity and the second equality by the above observation. $\qquad\square$

**Lemma 10.** *The data-dependent Shapley value satisfies the four data-dependent Shapley axioms.*

*Proof.* We show that each axiom is satisfied.

- **Balance:** By definition, $\sum_{i \in N} \phi_i^{\mathcal{D}} = \sum_{i \in N} \sum_{S:i \in S} \Pr\left[S \sim \mathcal{D}\right] C(S)/|S|$, then by switching the order of the summations,

$$\sum_{i \in N} \sum_{S:i \in S} \Pr\left[S \sim \mathcal{D}\right] \frac{C(S)}{|S|} = \sum_{S \subseteq N} \sum_{i \in S} \Pr\left[S \sim \mathcal{D}\right] \frac{C(S)}{|S|}$$
$$= \sum_{S \subseteq N} \Pr\left[S \sim \mathcal{D}\right] C(S)$$
$$= \operatorname*{E}_{S \sim \mathcal{D}}[C(S)].$$

- **Symmetry:** Let $\mathcal{S}_i = \{S : i \in S, \Pr\left[S \sim \mathcal{D}\right] > 0\}$. If $\Pr_{S \sim \mathcal{D}}\left[|S \cap \{i,j\}| = 1\right] = 0$, then $\mathcal{S}_i = \mathcal{S}_j$ and

$$\phi_i^{\mathcal{D}} = \sum_{S \in \mathcal{S}_i} \Pr\left[S \sim \mathcal{D}\right] \frac{C(S)}{|S|} = \sum_{S \in \mathcal{S}_j} \Pr\left[S \sim \mathcal{D}\right] \frac{C(S)}{|S|} = \phi_j^{\mathcal{D}}.$$

- **Zero element:** If $\Pr_{S \sim \mathcal{D}}\left[i \in S\right] = 0$, then $\Pr\left[S \sim \mathcal{D}\right] = 0$ if $i \in S$. Thus, $\phi_i^{\mathcal{D}} = 0$.

- **Additivity:** By definition of $\phi$ and $\alpha \mathcal{D}_1 + \beta \mathcal{D}_2$,

$$\phi_i^{\alpha \mathcal{D}_1 + \beta \mathcal{D}_2} = \sum_{S \,:\, i \in S} \Pr\left[S \sim \alpha \mathcal{D}_1 + \beta \mathcal{D}\right] \frac{C(S)}{|S|}$$
$$= \alpha \sum_{S:i \in S} \Pr\left[S \sim \mathcal{D}_1\right] \frac{C(S)}{|S|} + \beta \sum_{S:i \in S} \Pr\left[S \sim \mathcal{D}_2\right] \frac{C(S)}{|S|}$$
$$= \alpha \phi_i^{\mathcal{D}_1} + \beta \phi_i^{\mathcal{D}_2}.$$

$\square$

**Theorem 7.** *The empirical data-dependent Shapley value approximates the data-dependent Shapley value arbitrarily well, i.e.,*
$$|\tilde{\phi}_i^{\mathcal{D}} - \phi_i^{\mathcal{D}}| < \epsilon$$
*with* $\operatorname{poly}(n, 1/\epsilon, 1/\delta)$ *samples and with probability at least* $1 - \delta$ *for any* $\delta, \epsilon > 0$.

*Proof.* Define $X_j = \begin{cases} \frac{C(S_j)}{|S_j|} & \text{if } i \in S_j \\ 0 & \text{otherwise} \end{cases}$ and observe that $(\sum_{j=1}^m X_j)/m = \tilde{\phi}_i^{\mathcal{D}}$ and $\mathrm{E}[(\sum_{j=1}^m X_j)/m] = \phi_i^{\mathcal{D}}$. Clearly, $X_j \in [0, b]$ where $b := \max_S C(S)/|S|$, so by Hoeffding's inequality, $\Pr\left[|\tilde{\phi}_i^{\mathcal{D}} - \phi_i^{\mathcal{D}}| \geq \epsilon\right] \leq 2e^{-\frac{2m\epsilon^2}{\operatorname{poly}(n)}}$ with $0 < \epsilon < 1$. $\square$