[Reviews · NeurIPS 2017]

Reviewer 1



The paper studies the problem of statistical cost sharing in cooperative games where one does not observe the cost (or characteristic function) for every possible coalition. The goal is then to compute a cost sharing that approximate desired notions (core and Shapley value) while scaling well with the number of observed tuples (S, C(S)). The paper first gives results for the core: a bound to compute probably stable allocations where improves a previous bound (but is, according to the authors independently discovered in a paper currently on Arxiv), a better (logarithmic) bound for a weaker notion of approximate stability, and an impossibility result. Then, results for approximating the Shapley value are given and finally a new concept of statistical Shapley value is introduced that is easily approximable. Theorem 1 was, according to the authors, discovered independently from the reference Balcan et al 2016 currently on arxiv. It is of course impossible to check but this result is also only a small part of the overall contribution of this paper. I find the statistical cost sharing problem interesting and relevant and the results nice. I find the definition of the statistical Shapley value an interesting extension but I wonder how much sense it really makes in practice. It is basically tailored to be easy to approximate from the empirical average of costs observed from distribution \mathcal{D} but it is unclear to me that it still enjoy the nice properties of the classical Shapley value; for instance the fact of being thought as "fair" or being stable / giving incentives to users to join, etc. Minor comments: - typo in abbstract, remove curvature once - In the definition of the statistical cost sharing problem: is N necessarily one of the observed costs in the sequence?

Reviewer 2



This paper continues the work of Balcan et al. ’15 on learning cost sharing functions in cooperative game theory from sample observations. Classical work in cooperative game theory considers the existence and knowledge of a cost function C(.) such that C(S) is the cost of a set S. Works in this space are interested in computing cost-sharing values \phi_i for all individuals i, that have certain properties. The first notion of interest discussed in this work is the Core. Core is a set of values \phi_i that incentivizes all individuals to form one coalition. The second cost-sharing notion is the Shapley value, that at a high level coincides with the marginal cost of a player in a random ordering of all players. Both of these concepts can be defined by properties that are discussed in section 2.1. In this work, the authors consider a setting where the cost function C(.) is not known in advance, but samples of the form (S, C(S)) are drawn from some distribution over sets. The goal of the paper is to use these samples to learn values \phi_i that (approximately) preserve the properties associated with the definition of Core and Shapely value on sets S that are drawn from the same distribution. I like the general premise of the paper. I agree that many game theoretic models rely on the type of knowledge that might not be available, in the case of this paper the ability to know C(.) fully. So, I find it valuable to explore how one can use a learning theoretic framework for creating robust versions of these game theoretic definitions that can be learned from an environment. This is not a new direction, however, as it has been explored by Balcan et al’15 for cooperative games, and a large body of work for other aspects of game theory, such as social choice, and mechanism design. One concern is that a number of the results in this paper are either marginal improvements over previous works, or are results that are said to be “independently discovered by Balcan et al.’16”. I find this mention of “independently” a little awkward, given that the work has been published. I would normally not hold this against a result, specially if the result requires deep insights where independent work sheds new light. But, as far as machine learning insight go in these example, the methodology is quite simple — noticing that the properties describing a notion create an LP whose constrains are given by the samples and therefore can be learned. There are also novel directions in this work that haven’t been explored in previous works. For example, the authors also explore how to approximate the Shapley value and also introduce a variation of shapley value that satisfies probabilistic (data-dependent) properties. All together, I like the paper and the direction it’s taking. But, I’m not entirely convinced that the contributions of this paper constitute new interesting results that would be of special interest the community. After rebuttal: I have read the author's response.

Reviewer 3



This paper studies cost sharing in cooperative games. There is a ground set N of players and a cost function C that maps subsets S of N to a real-valued cost C(S). The goal is to split the entire cost of the ground set C(N) among the players in such a way that certain desirable properties are satisfied, such as the “core property” (defined in Section 2.1) or the “Shapley axioms” (defined in Section 2.2). Prior work in the cost sharing literature has designed algorithms computing the cost shares which have oracle access to the underlying cost function C. Recent work by Balcan, Procaccia, and Zick in IJCAI 2015 studies the learning theoretic setting where the algorithm does not have oracle access but receives a sample of subsets labeled by their cost {(S1, C(S1)), (S2, C(S2)), …, (Sm, C(Sm))}. In this statistical cost sharing model, the authors of the present paper design sample efficient and computationally efficient learning algorithms that compute cost shares that (approximately) satisfy various properties with high probability. From what I understand, in Balcan et al.’s IJCAI 2015 paper, they showed that for some games, it is possible to efficiently compute a “probably approximately stable core” (Definition 3) using a polynomial number of samples and then in the arXiv paper published a year later they showed that this is true for all games with a non-empty core. This matches Theorem 1 of the present paper. The present paper builds on Balcan et al.’s work by defining two other notions of an approximate core (Definition 3). They prove various upper and lower bounds on sample complexity and computability for these other notions (Theorems 2 and 3). Finally, the authors study cost shares that approximate the “Shapley value” which is a solution to the cost sharing problem satisfying several desirable properties. They show that if the cost function is a submodular function, then the extent to which the Shapley value can be approximated from samples from the uniform distribution/any bounded product distribution depends on the submodular function’s curvature (with upper and lower bounds). The also show that if the cost function is a coverage function, then it can’t be approximated from samples over the uniform distribution. Overall, I think this paper makes a solid contribution to an interesting and applicable subject area. There are a few obvious ways the theorems could be improved: - Theorem 2: It’s too bad that the sample complexity depends on the spread of C since it seems reasonable that the spread of C might be comparable to n in many settings. Whether or not the dependence can be removed is especially interesting since Theorem 1 has no dependence on the spread. Do the authors have any intuition about whether or not the dependence on the spread is necessary? - Theorem 4: Can this theorem be generalized to other distributions? If the authors have any interesting insights into the key technical challenge here, that would be a nice addition to the discussion. I think the axioms for the data-dependent Shapley value could be better motivated. Specifically, the symmetry and zero element axioms seem a bit unintuitive because there’s no connection to the cost function. To me, a more natural distributional parallel to the original symmetry condition would be something like, “For all i and j, if Pr_{S \sim D}[C(S \cup {i}) = C(S \cup {j})] >= ½, then \phi_i^D = \phi_j^D” (I chose ½ arbitrarily). Also, in my opinion, a more natural distributional parallel for the original zero element condition could be “For all i, if Pr_{S \sim D}[C(S \cup {i}) – C(S) = 0] >= ½, then \phi_i^D = 0.” Small questions/comments: - I think Theorem 1 should say “probably approximately stable core.” - In the proof sketch of Theorem 4, what is C_{A_{sigma < i}}(i)? =======After author feedback======= Thanks, it would be great to include examples where a dependence on the spread is necessary. And I see, that's a fair point about how you have limited latitude in crafting alternative characterizations.